# Comparison of the Chemical Properties of Pineapple Vinegar and Mixed Pineapple and Dragon Fruit Vinegar

Antika Boondaeng [1], Sumaporn Kasemsumran [1], Kraireuk Ngowsuwan [1], Pilanee Vaithanomsat [1], Waraporn Apiwatanapiwat [1], Chanaporn Trakunjae [1,2], Phornphimon Janchai [1], Sunee Jungtheerapanich [1] and Nanthavut Niyomvong [3,*]

1 Kasetsart Agricultural and Agro-Industrial Product Improvement Institute, Kasetsart University, Bangkok 10900, Thailand
2 School of Biological Sciences, University Sains Malaysia, George Town 11800, Penang, Malaysia
3 Department of Biology and Biotechnology, Faculty of Science and Technology, Nakhon Sawan Rajabhat University, Nakhon Sawan 60000, Thailand
* Correspondence: nanthavut.ni@nsru.ac.th; Tel.: +66-09-6960-6769

**Abstract:** Pineapples are a tropical fruit with high nutritional value and high vitamin and sugar contents. In this study, low-grade pineapples were fermented to produce vinegar using surface culture fermentation (SCF), which involved the addition of dragon fruit juice, to compare the quality and antioxidant activity of different preparations of vinegar. The highest acetic acid concentration (7.35%) was obtained from pineapple vinegar after 20 days of incubation. Vinegar made from mixed pineapple and dragon fruit juice without peel and vinegar with pineapple and dragon fruit juice with peel had acetic acid concentrations of up to 6.20% and 4.50%, respectively. The mixed-fruit vinegar of pineapple and dragon fruit juice with peel displayed the highest antioxidant activity at 210.74 μg/g TE, while no significant difference was found between the other two vinegars (189.52 vs. 187.91 μg/L TE). Notably, the volatile compounds detected in the vinegars were alcohols and esters, which may contribute to the distinct aroma. Overall, the addition of dragon fruit juice with peel to pineapple vinegar increased the phenolic content and antioxidant activity; however, fermentation was slightly slower than that of the other two test materials.

**Keywords:** dragon fruit; pineapple; mixed-fruit vinegar; *Acetobacter aceti*; surface culture fermentation

## 1. Introduction

Thailand has many types of fruit that can be used as raw materials for fermenting vinegar for added value, such as okra, mulberries, mangos, mangosteens, dragon fruit, and pineapples. The important pineapple planting areas in Thailand include Uthai Thani, Prachuap Khiri Khan, Rayong, Ratchaburi, Chon Buri, Phitsanulok, and Phetchaburi. Pineapple prices have declined owing to their oversupply. Accordingly, repurposing over-supplied pineapples for added product value (i.e., synthesizing products, such as pineapple wine, pineapple cider, or pineapple vinegar) might increase income and reduce costs.

In addition to pineapple, dragon fruit is a popular fruit that is widely cultivated in Asia. Dragon fruit has a sweet taste and high nutritional value; however, the supply of this fruit sometimes exceeds the needs of consumers. As a result, dragon fruit has been used to formulate many products. Dragon fruit contains many antioxidants, including betalains (in red dragon fruit) that help prevent bad cholesterol (LDL), hydroxycinnamate, which has anti-cancer effects, and flavonoid antioxidants that possess brain-nourishing properties and reduce the risk of heart disease [1–3]. Although dragon fruit is rich in various nutrients and fibers, it has a sweet taste that lacks uniqueness. Compared with other fruits, dragon fruit has a sugar content that exceeds 10 °Brix, which is not suitable for individuals attempting to control their sugar intake. If the amount of sugar can be reduced via the addition of a unique flavor while nutritional value and color are maintained, dragon fruit will have

added value [4]. The other approach involves processing fruits into healthy beverages, such as fermented vinegar using yeast and bacteria via fermentation with pineapple juice to achieve more unique flavor combinations.

Vinegar is classified as a food; thus, its quality or standard is determined according to the "Ministry of Public Health (No. 204) B.E. 2000" [5]. Vinegar is obtained by fermenting grains, fruits, sugar, or molasses, such as rice, corn, pineapples, apples, oranges, strawberries, mangoes, and bengal currant. Sugary raw materials directly feed yeast; however, starchy ingredients, such as rice and glutinous rice, must be converted to sugar before fermentation into vinegar via the two-step fermentation method. This method involves alcoholic fermentation using yeast, followed by acetic acid fermentation with *Acetobacter* and *Gluconobacter* bacteria in the presence of oxygen. Fermented vinegar has a clear color with only natural sediment, a good smell that depends on the smell of raw materials, and a good taste. The acetic acid content is not less than 4%, and the residual alcohol content is not more than 0.5% [5].

Nowadays, fermented vinegar has become popular as a healthy food as it is rich in minerals, vitamins, and prebiotics. Prebiotics are foods that cannot be digested by the human body and are not absorbed in the gastrointestinal tract via the stomach and small intestine. However, prebiotics are digested by bacteria in the colon. Bacteria stimulate their activity and produce probiotics, which are classified as a functional food group with many benefits. These benefits include enhancing the absorption of minerals, developing a strong immune system, stimulating the activity of macrophages, improving the immune environment of the digestive tract, preventing gastrointestinal infections and diarrhea, increasing beneficial microorganisms, inhibiting pathogenic microorganisms, reducing the risk of intestinal infections, absorbing toxins in the digestive tract, softening the stool, enabling easy excretion, and balancing the digestive system to prevent colon cancer. To maintain intestinal balance, prebiotics or foods that help nourish microorganisms in the intestines should be consumed, which ultimately provides another approach to help strengthen the body [6–8].

In Thailand, vinegar is produced from various agricultural raw materials, such as Leum Pua glutinous rice, using enzymes for raw starch digestion [9,10] and surface culture fermentation (SCF) with *A. aceti* TISTR 354 at 30 °C. After 6 days of fermentation, the acetic acid content increases by 5.7%, and antioxidant activity is observed. Saithong et al. [10] reported the fermentation of Nipa sap with mixed yeast *Saccharomyces cerevisiae* prior to acetic acid fermentation by *A. aceti* TISTR 354 using SCF. SCF led to a 2.9-fold (6.2%) increase in acetic acid compared with the conventional method (2.14%). Furthermore, the antioxidant activity and total phenolic compounds in the product were found to be higher than those in commercial vinegar.

Vinegar fermented from local fruits, such as pineapple and dragon fruit, is another interesting option owing to its economic advantages and high content of substances that are beneficial to the body. In this study, pineapple juice was fermented with dragon fruit to create unique flavor combinations, increase its antioxidant activity, and create appealing colors that will attract more consumers.

## 2. Materials and Methods

### 2.1. Preparation of the Fruit Juices

Ripe low-grade pineapples (*Ananas comosus* L. Merr c.v. Patavia) and red dragon fruit (*Hylocereus polyrhizus*) were purchased from a local market in Bangkok, Thailand. The fruit samples were cleaned with tap water, peeled, and homogenized in a blender to obtain their respective fruit juices. The chemical characteristics of the fruit samples, such as their total soluble solids (TSS), reducing sugar, total titratable acid (TTA) content, pH, total phenolic compounds (TPCs), and antioxidant activity were determined.

### 2.2. Alcoholic Fermentation

The appropriate ratio of pineapple juice to water used in this study was obtained from a previous study [11]. Three ratios of pineapple juice to water and red dragon fruit juice (Table 1) were mixed to obtain three treatments: T1, T2, and T3. The total soluble solids (TSS) and pH were adjusted to 25 °Brix and 3.5–4.0 using sucrose and baking soda or citric acid, respectively. The treatments were decontaminated via addition of potassium metabisulphite ($K_2S_2O_5$) to achieve a final concentration of 75–100 mg/L. An inoculum of 5% *v/v* (*S. cerevisiae* var. burgundy), purchased from the Department of Applied Microbiology, Institute of Food Research and Product Development (IFRPD), Kasetsart University, Thailand, was transferred to all treatments, which were then fermented at 25 °C. The fermented sample juices were collected for microbiological and chemical analyses. When the alcohol concentration reached 10.0% (*v/v*), fermentation was terminated. The fruit wines were stored at 4 °C for use as raw material for vinegar fermentation.

**Table 1.** Ratios of pineapple juice to water and pineapple juice to red dragon fruit juice.

| Treatment | Ratios of Juices and Water | | | |
| | Pineapple Juice | Red Dragon Fruit Juice | Red Dragon Fruit Peel Juice | Water |
| --- | --- | --- | --- | --- |
| T1 | 4 | 0 | 0 | 2 |
| T2 | 2 | 1 | 0 | 3 |
| T3 | 2 | 0.5 | 0.5 | 3 |

### 2.3. Vinegar Fermentation

The starter culture was prepared using *A. aceti* TISTR 354 purchased from the Department of Applied Microbiology, Institute of Food Research and Product Development (IFRPD), Kasetsart University, Thailand. The culture was prepared in 90 mL of sterilized juice with an initial sugar concentration of 5 °Brix, 3 mL of 95.0% ethanol, and 7 mL of *A. aceti*, and was incubated at 30 °C for 72 h before use. Vinegar fermentation was performed using SCF [12]. SCF involved two steps. First, fermentation was initiated by mixing the sterilized juice, wine, and starter culture at a ratio of 600:300:100 and incubating the mixture at 30 °C for 48 h. Thereafter, 1000 mL of each wine was added to the fermentation broth, which was then incubated for 18 days. Finally, the samples were collected for microbiological and chemical analyses. All experiments were conducted in triplicate.

### 2.4. Chemical Analysis

A pH meter (Model PH1200, Horiba, Japan) and a hand refractometer (RHB-32ATC, Shenzhen City, China) were used to measure the pH and TSS, respectively; TSS was reported as °Brix for soluble solid content. The TTA [13] and volatile acidity (VA) [14] were determined using acetic acid and citric acid, respectively, via titration with 0.1 N NaOH using phenolphthalein as an indicator. The reducing sugar content was estimated using a Nelson-Somogyi assay [15].

The ethanol concentration was assessed using gas chromatography (Chromosorb-103, GC4000; GL Sciences; Tokyo, Japan) with an HP5 capillary (30 m × 0.32 mm × 0.25 μm; JW Scientific, Folsom, CA, USA) [16]. The viability of microorganisms was determined using a spread plate. Serial dilutions were spread on yeast extract calcium carbonate agar (GYC agar) and incubated at 28 °C for 1–2 d. This analysis was performed in triplicate. The microorganism colonies were counted to obtain the population in log CFU/mL [17].

The acetic acid content was determined using high-pressure liquid chromatography (HPLC) with a Bio-Rad Aminex HPX-87H column (300 × 7.8 mm Bio-Rad Laboratories Inc., Hercules, CA, USA) and a Shimadzu RID-UV detector. The mobile phase comprised 5 mM $H_2SO_4$ at a flow rate of 0.6 mL/min and temperature of 60 °C. Samples were filtered through a 0.25 mm microporous membrane filter prior to HPLC analysis. A standard solution of acetic acid with 99.8% purity (Sigma-Aldrich, St. Louis, MO, USA) was prepared for the

HPLC calibration curve [18]. The color of the fruit vinegars was evaluated using MiniScan EZ (MSEZ1949, HunterLab, Reston, VA, USA), with color values of L*, a*, and b* [19].

### 2.5. Determination of Total Phenolic Compounds

The Folin-Ciocalteu colorimetric method [20] was used to determine the TPC (total phenolic compound) content in the vinegars. Briefly, 0.3 mL of each vinegar sample was mixed with Folin-Ciocalteu reagent (1.5 mL). After 5 min, 1.2 mL of 7.5% (*w/v*) sodium carbonate solution was added to the mixture, which was then kept in the dark for 30 min at room temperature. The absorbance was measured at 765 nm using a spectrophotometer (Thermo Fisher Scientific 4001/4 Genesys 20; Waltham, MA, USA). Gallic acid was used as the standard. TPC is expressed as mg gallic acid equivalent per ml of vinegar (mg GAE/mL).

### 2.6. Determination of Antioxidant Activity

The free radical scavenging activity of vinegar was estimated using a diphenyl-p-picrylhydrazyl (DPPH) radical scavenging capacity assay [21]. The absorbance was measured spectrophotometrically (Shimadzu UVmini-1240, Kyoto, Japan) at 517 nm after a 30 min reaction in the dark. The results are expressed in µg Trolox equivalent (µg TE/g) of vinegar.

### 2.7. Fourier Transform Infrared (FTIR) Analysis

A Nicolet IR200 FTIR spectrometer (Thermo Scientific, Madison, WI, USA) was employed for Fourier-transform infrared spectroscopic (FTIR) analysis. The spectra were recorded in the range 500–4000 $cm^{-1}$ with a mean of 32 scans and a resolution of 4 $cm^{-1}$. The FTIR spectra were plotted as intensity versus wave number [22].

### 2.8. Gas Chromatography-Mass Spectrometer Analysis

Qualification of the volatile compounds was performed using GC-MS (Shimadzu, Nexis GC-2030NX, Japan) and a DB-5MS column (30 m × 0.32 mm, 0.5 µm; Agilent, USA) [23]. The samples were incubated in headspace vials at 60 °C for 20 min. The injection was conducted in splitless mode at 280 °C. Chromatographic separation was conducted using the following program: 80 °C for 5 min, increased to 150 °C at a rate of 5 °C/min, and increased to 280 °C at a rate of 10 °C/min. Helium was used as the carrier gas at a constant flow rate of 1.49 mL/min. The mass spectrometer was operated at a transfer line temperature, and the ion source was operated at 160 °C and 280 °C. The volatile compounds in the vinegars were identified by comparing their relative retention times and mass fragmentation patterns with the data system library (Wiley 7 NIST 12 and NIST 62) and retention data of commercially available standards.

### 2.9. Statistical Analysis

Treatment differentiation was evaluated using statistical analysis of variance followed by Duncan's multiple range test using SPSS Software v. 20.0 (IBM Analytics, USA). The means were considered significantly different at $p < 0.05$.

## 3. Results and Discussion

### 3.1. Chemical Characteristics

The chemical characteristics of the pineapples and red dragon fruits obtained from a local market in Bangkok, Thailand, are listed in Table 2. The TSS of pineapple juice was slightly higher (13.07 °Brix) than that of red dragon fruit juice (11.93 °Brix), while its TTA was higher than that of red dragon fruit juice, resulting in a lower pH. Pineapple juice had relatively higher reducing sugar and nitrogen contents than red dragon fruit juice. However, the nitrogen content of both juices was >0.025 g/L *w/v*, which is sufficient for yeast growth [24]. Using a DPPH radical scavenging assay, pineapple juice was found to possess total phenolics and total antioxidant activities of 407.00 mg/L GAE and 198.29 µg/g

TE, respectively, which were higher than those of red dragon fruit juice and red dragon fruit peel. Owing to the total concentrations, especially the nitrogen content, pineapple juice can be used as a raw material for vinegar fermentation.

**Table 2.** Basic chemical characteristics of pineapple and dragon fruit.

| Chemical Characteristics | Value ± SD | | |
|---|---|---|---|
| | Pineapple | Dragon Fruit | Dragon Fruit Peel |
| Total soluble solid (TSS, °Brix) | 13.07 ± 0.12 | 11.93 ± 0.23 | - |
| Reducing sugar (g/L) | 92.13 ± 1.34 | 48.24 ± 2.42 | - |
| pH | 3.58 ± 0.02 | 4.44 ± 0.01 | - |
| Total titratable acidity (%) | 0.286 ± 0.00 | 0.124 ± 0.00 | - |
| Antioxidant activity (µg TE/g) | 198.29 ± 1.51 | 162.60 ± 8.13 | 123.24 ± 0.82 |
| Total phenolic compound (mg GAE/L) | 407.00 ± 8.60 | 256.50 ± 0.58 | 134.95 ± 7.07 |
| Nitrogen content (*w/v*%) | 0.08 ± 0.01 | 0.027 ± 0.20 | - |

*3.2. Alcoholic Fermentation*

Three ratios of pineapple juice to water and dragon fruit juice (Table 1) were used throughout the fermentation with *S. cerevisiae* var. burgundy. During fermentation, the pH value decreased from the initial value of 4.0 to 3.50, while the TTA content increased from fermentation day 1 to day 10 (Figure 1). The change in the TTA content in treatment T1 was significantly different from that in the other treatments. During alcoholic fermentation, other substances, including acetic acid, glycerol, and higher alcohols, were produced, resulting in a lower pH and higher TTA. The increase in alcohol content in all treatments corresponded to a decrease in the TSS. The highest alcohol concentration obtained using treatment T1 (10.71% *v/v*) was not significantly different from that obtained using treatments T2 (10.07 *v/v*) and T3 (10.22% *v/v*) (Figure 2). At the end of fermentation, the alcohol content did not increase owing to the absence of fermentable sugars for alcohol conversion. Further, the alcohol content insignificantly decreased owing to self-evaporation and yeast metabolism [25].

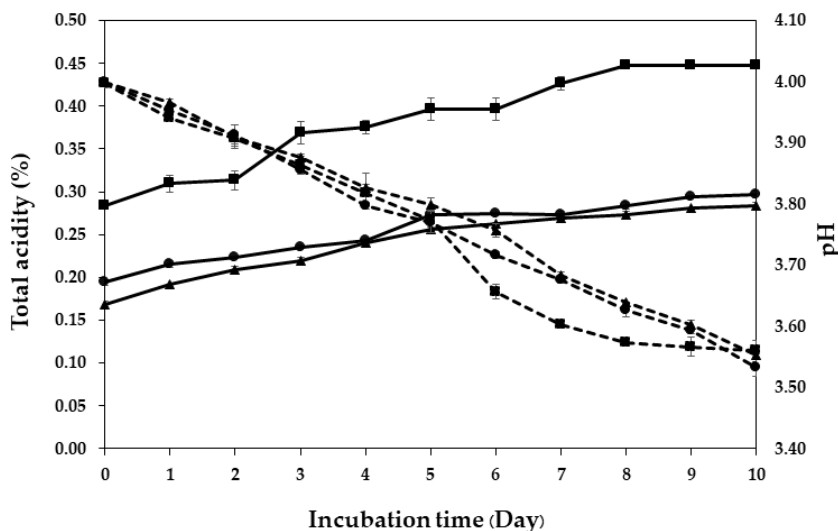

**Figure 1.** Total acidity (%) (solid line) and pH (dashed line) of pineapple wine (■), pineapple wine mixed with dragon fruit without peel (▲), and pineapple wine mixed with dragon fruit with peel (●) during wine fermentation.

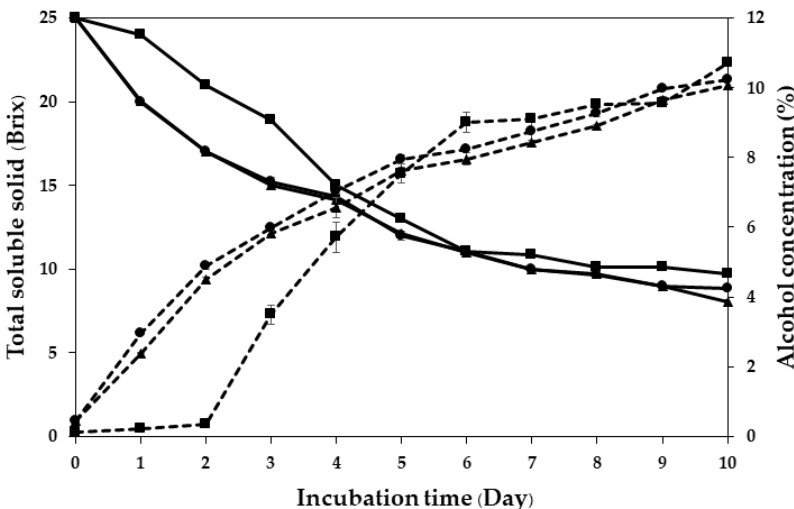

**Figure 2.** Total soluble solids (solid line) and alcohol content (dashed line) of pineapple wine (■), pineapple wine mixed with dragon fruit without peel (▲), and pineapple wine mixed with dragon fruit with peel (●) during alcoholic fermentation.

### 3.3. Vinegar Fermentation

Fruit vinegar was obtained from the fermentation of *A. aceti* TISTR 354 using the fruit wines derived above as substrates. After 48 h of cultivation, $2.5 \times 10^6$ CFU/mL of bacterial cells in 100 mL of starter culture was mixed with 600 mL of each fruit juice (5 °Brix) and 300 mL of each fruit wine. In the first step of the SCF process, 1000 mL of the mixture was incubated for 48 h at room temperature. Based on the results, the acetic acid content in pineapple vinegar, pineapple vinegar mixed with red dragon fruit juice without peel, and pineapple vinegar mixed with red dragon fruit juice with peel reached 2.13%, 1.52%, and 1.97%, respectively, within 2 days (Figure 3a). The alcohol content in the fermentation broth was oxidized by *A. aceti* TISTR 354 under aerobic conditions to produce acetic acid [26], aligning with the decrease in pH (Figure 3b). To improve the efficiency of acetic acid production, 1000 mL of each fruit wine was added to the vinegar fermentation broth, which was the second step of SCF. After the addition of each fruit wine, the ethanol concentration increased, whereas the acetic acid concentration decreased. The increased ethanol content resulted in an increase in acetic acid content, which served as the final product of the process from the oxidation of *A. aceti* TISTR 354, with ethanol as the substrate. Eighteen days after the addition of each fruit wine, the acetic acid concentrations in pineapple vinegar, pineapple vinegar mixed with red dragon fruit juice without peel, and pineapple vinegar mixed with red dragon fruit juice with peel increased from 1.41 to 7.35% *v/v*, 1.23% to 6.20%, and 1.16% to 4.50%, respectively, while the residual alcohol concentrations were 0.06%, 1.15%, and 1.85%, respectively. The pH of all treatments decreased from 3.92–4.10 to 2.80–2.90 during vinegar fermentation using SCF (Figure 3b). Theoretically, 1 g of ethanol can be converted to 1.3 g of acetic acid during vinegar production; however, in practice, the yield obtained is approximately 15–20% *v/v* lower because alcohol, acetaldehyde, and acetic acid are volatile [27]. The pattern of bacterial populations is shown in Figure 3c. During day 10, the acetic acid bacteria population increased approximately 2 log cycles in pineapple vinegar and pineapple vinegar mixed with red dragon fruit juice without peel, and increased 1 log cycle in pineapple vinegar mixed with red dragon fruit juice with peel. After that, the yeast went through an early death phase, decreasing approximately 4 log cycles through to the last day.

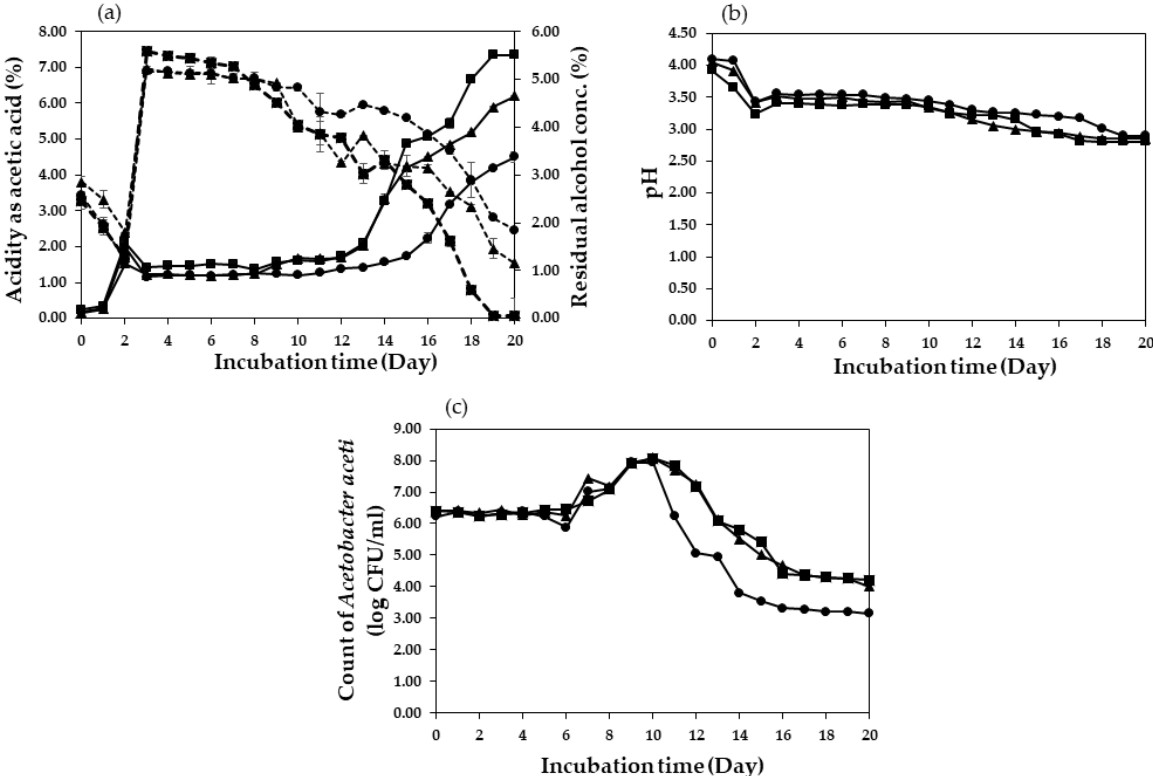

**Figure 3.** The change in acetic acid (solid line) and residual alcohol (dash line) concentrations (**a**), the pH values (**b**), and bacterial population (**c**) of pineapple vinegar (■), pineapple vinegar mixed with dragon fruit without peel (▲), and pineapple vinegar mixed with dragon fruit with peel (●) during vinegar fermentation by *A. aceti* TISTR 354.

According to the "Ministry of Public Health (No. 204) B.E. 2000", vinegar should not contain less than 4% acetic acid and not more than 0.5% alcohol residue. However, according to the US Food and Drug Administration, the acetic acid content of vinegar should be at least 4% [28]. The EU states that fermented vinegar should contain at least 5% acetic acid and have a maximum alcohol content of 0.5%. Wine vinegar obtained from acetic acid fermentation should contain acetic acid content not less than 6% *w/v*, and the residual alcohol content should not exceed 1.5% v/g ("Regulation (EC) No. 1493/1999"). In this study, pineapple vinegar met the above standards, including the properties of wine vinegar. Pineapple vinegar mixed with dragon fruit without peel had wine vinegar properties, and pineapple vinegar mixed with dragon fruit with peel contained the standard acetic acid content; however, the residual alcohol exceeded 0.5%. Thus, the fermentation time should be required to enable oxidization of the alcohol to acetic acid. Additionally, aeration may be required for better fermentation.

Compared to previous research, Tanamool et al. [29] revealed that pineapple vinegar produced the highest amount of acetic acid (6.5% *w/v*) in 25 days under simultaneous fermentation conditions. Further, Sossou et al. [30] reported that vinegar produced from pineapple peels had 4.5% *w/v* acetic acid following 23–25 days of sequential fermentation. Based on these experimental data, SCF and *A. aceti* TISTR 354 were determined to be effective for producing high-quality mixed-fruit vinegar.

### 3.4. Total Phenolic Content and Antioxidant Activity

The total phenolic content in all trials was found to be relatively stable (Table 3), with a slight decrease in pineapple vinegar from the first day (282.74 mg GAE/L) to the last day (245.31 mg GAE/L). In contrast, the antioxidant content increased from the first day to the last day (from 143.76 μg/g TE to 189.52 μg TE/g).

**Table 3.** Total phenolic compound and antioxidant activity of the fruit vinegar products.

| Samples | Total Phenolic (µg GAE/mL) | | Antioxidant DPPH Assay (µg TE/g) | |
|---|---|---|---|---|
| | Day 0 | Day 20 [ns] | Day 0 | Day 20 |
| Pineapple vinegar | 282.74 [a] ± 4.32 | 245.31 ± 2.88 | 143.79 [b] ± 0.43 | 189.52 [b] ± 1.53 |
| Pineapple vinegar mixed with red dragon fruit without peel | 121.11 [c] ± 3.96 | 234.63 ± 6.48 | 156.88 [a] ± 3.07 | 187.91 [b] ± 1.31 |
| Pineapple vinegar mixed with red dragon fruit with peel | 186.41 [b] ± 12.8 | 259.97 ± 9.09 | 131.22 [c] ± 3.97 | 210.74 [a] ± 1.61 |

[a], [b], [c], means with different letters in the same row of each kinetic parameter are significant at $p \leq 0.05$. [ns], means in the same row are not significant at $p > 0.05$.

The phenolic content of pineapple vinegar with dragon fruit without peel increased from 121.11 mg GAE/L to 189.74 mg GAE/L, which was similar to the antioxidant activity, which increased from 156.88 µg TE/g to 187.91 µg TE/g.

Pineapple vinegar and pineapple vinegar mixed with dragon fruit without peel did not have statistically different antioxidant activities, whereas vinegar mixed with dragon fruit with peel had the highest antioxidant activity.

The phenolic content of pineapple vinegar tended to increase from the first day of fermentation to the last day, in contrast to the antioxidant activity. Normally, phenolic content analysis measures the reducing capacity in the same manner as antioxidant activity analysis; therefore, a linear relationship or the same direction has been found between antioxidant activity and phenolic content in most experiments [31]. However, the increased phenolic content, contrary to the antioxidant activity in this experiment, may be due to the sulfur and sugar in pineapple juice, which exaggerates the phenolic value, as sulfur dioxide and sugar interfere with the Folin-Ciocalteu reagent [32]. The analysis of phenolic content with the Folin-Ciocalteu reagent uses the principle of phenolic dissociation into protons and anions of phenolate, which reduces the Folin-Ciocalteu reagent and results in a blue solution. Other substances without phenolics, such as sugar, aromatic amines, ascorbic acid, and organic acids, can be reduced by the Folin-Ciocalteu reagent, resulting in a blue solution and a high phenolic content reading.

The colors of all three types of vinegar were measured in terms of brightness ($L^* = 0$ black and $L^* = 100$ colorless), green ($-a^*$) or red ($a^*$), and blue ($-b$) or yellow ($b^*$). Pineapple vinegar had a yellowish transparent color with $L^*$ ($13.98 \pm 1.03$), $-a^*$ ($-1.18 \pm 0.07$), and $b^*$ ($5.57 \pm 0.7$) values. The color of pineapple vinegar mixed with dragon fruit without peel was clear orange, with $L^*$, $a^*$, and $b^*$ values of $5.34 \pm 0.29$, $1.88 \pm 0.13$, and $2.08 \pm 0.31$, respectively. Pineapple vinegar mixed with dragon fruit and peel had an orange-pink color, with $L^*$, $a^*$, and $b^*$ values of $6.50 \pm 0.22$, $3.97 \pm 0.25$, and $2.13 \pm 0.31$, respectively.

Preliminary chemical composition analysis of the vinegars by gas chromatography-mass spectrometry (GC-MS) revealed that all vinegars contained six identical volatile substances: ethanol, acetic acid, acetic acid ethyl ester, acetic acid formyl ester, isopentyl alcohol, 1-butanol, 3-methyl, and acetate (Table 4). Pineapple vinegar also had two additional volatile substances, isobutyl acetate and 2,3-butanediol, while pineapple vinegar mixed with dragon fruit without peel contained isobutyl acetate. Pineapple vinegar mixed with dragon fruit with peel did not contain isobutyl acetate or 2,3-butanediol. It can be seen that acetic acid ethyl ester (also ethyl acetate or vinegar ester) was the primary compound in these vinegars. This is in accord with the results obtained by Callejón et al. [33] and Plioni et al. [34], who reported that acetic acid ethyl ester is usually found in higher proportions in the volatile fraction than in other compounds, and that it is also important for vinegar's sensory characteristics. These results imply that volatile substances affect the aroma and quality of vinegar; however, each aroma differs according to the raw materials used.

**Table 4.** Volatile compounds identified in pineapple vinegar samples using gas chromatography-mass spectrometer (GC-MS).

| Assignment Compounds | Retention Time | | | % Area | | | % Similarity | | |
|---|---|---|---|---|---|---|---|---|---|
| | **T1** | **T2** | **T3** | **T1** | **T2** | **T3** | **T1** | **T2** | **T3** |
| Ethanol | 1.314 | 1.309 | 1.304 | 15.53 ± 0.76 | 19.75 ± 0.16 | 25.24 ± 0.73 | 98.00 ± 0.00 | 98.00 ± 0.00 | 98.00 ± 0.00 |
| Acetic acid | 1.458 | 1.448 | 1.434 | 30.50 ± 0.10 | 28.18 ± 0.14 | 25.73 ± 0.14 | 96.00 ± 0.00 | 96.00 ± 0.00 | 96.00 ± 0.00 |
| Acetic acid ethyl ester | 1.538 | 1.533 | 1.533 | 46.05 ± 0.76 | 46.42 ± 0.10 | 41.78 ± 0.99 | 96.00 ± 0.00 | 96.00 ± 0.00 | 96.00 ± 0.00 |
| Acetic acid formyl ester | 1.887 | 1.885 | 1.885 | 0.25 ± 0.01 | 0.30 ± 0.01 | 0.24 ± 0.11 | 88.00 ± 0.00 | 88.00 ± 0.00 | 90.67 ± 4.62 |
| Isopentyl alcohol | 2.028 | 2.021 | 2.020 | 5.96 ± 0.14 | 4.54 ± 0.06 | 6.86 ± 0.35 | 96.00 ± 0.00 | 96.00 ± 0.00 | 97.00 ± 1.73 |
| Isobutyl acetate | 2.280 | 2.262 | - | 0.30 ± 0.02 | 0.22 ± 0.01 | - | 97.00 ± 0.00 | 97.00 ± 0.00 | - |
| 2,3-Butanediol | 2.345 | - | - | 0.04 ± 0.08 | 0.13 ± 0.00 | - | 93.00 ± 3.47 | - | - |
| 1-Butanol, 3-methyl-, acetate | 3.536 | 3.527 | 3.524 | 1.35 ± 0.03 | 0.60 ± 0.01 | 0.23 ± 0.02 | 97.00 ± 0.00 | 99.00 ± 0.00 | 99.00 ± 0.00 |

T1 = pineapple vinegar. T2 = pineapple vinegar mixed with red dragon fruit without peel. T3 = pineapple vinegar mixed with red dragon fruit with peel.

When vinegar was analyzed by FTIR, the peaks in the IR spectrum displayed functional group oscillations corresponding to various biomolecules. Figure 4 shows the vinegar spectrum in the wavelength of 4000–500 cm$^{-1}$. In general, red dragon fruit contains betalains, flavonoids, phenolic acids, phenylpropanoids, terpenes, steroids, polysaccharides, and fatty acids [25]. According to Barkociová et al. [35], the FTIR spectrum of dragon fruit revealed vibration of the C-O stretching group at 850–897 cm$^{-1}$, vibration of the C=N groups at 1590–1634 cm$^{-1}$, and vibration of the C-N groups at 1031–1078 cm$^{-1}$. The wavelengths of 1403–1417 cm$^{-1}$ and 920–929 cm$^{-1}$ indicate vibrations of the -OH bonds [35]. These findings indicate that pineapple vinegar mixed with dragon fruit with/without peel analyzed by FTIR had peaks at 850–897 cm$^{-1}$, 1031–1078 cm$^{-1}$, and 1403–1417 cm$^{-1}$, which imply that the vibrations of the C-O, C=N, and C-N stretching groups were not found in pineapple vinegar. The main components of these vinegars were consistent with those found in previous research, which revealed wavelengths of 3800–2790 cm$^{-1}$ and 1685–1550 cm$^{-1}$ for the vibrating ranges of the -OH group of water and C-H stretching of acetic acid during vinegar composition analysis. The wavelength range of 1300–1000 cm$^{-1}$ is the vibration range of organic acid, whereas 1100–1000 cm$^{-1}$ represents the C-O stretching vibration of ethanol. The peaks found in the 1700–1600 range indicate the vibrating range of the CO stretching functional group in the structure of aldehyde compounds, while the range of 1800–900 cm$^{-1}$ indicates the -C-O and -OH groups that vibrate in phenolic compounds [36,37].

According to a previous study, pineapple vinegar mixed with dragon fruit and peel has the highest antioxidant activity. However, the amount of acetic acid produced is less than that of pineapple vinegar and pineapple vinegar mixed with dragon fruit without peel, which has antioxidant properties related to the phenolic content. Of note, phenolic compounds are a group of substances with antioxidant properties. The antioxidant mechanism involves the addition of hydrogen atoms to free radicals, leading to termination of the chain reaction. DPPH radicals can be used as a model to study the antioxidant activity of a substance and is a faster approach than other methods. The antioxidant activity of wine against DPPH radicals is due to the presence of hydrogen atoms [38]. When DPPH radicals accept electrons, they become stable [39]; however, the phenolic content and antioxidant capacity depend on the type of fruit and the production of vinegar.

When the pineapple vinegar produced in this study was compared with commercial pineapple vinegar, all fermented vinegars were found to have a higher total phenolic content than commercial pineapple vinegar, and it may have had higher antioxidant activity, as a linear relationship or at least positive correlation has been found for antioxidant activity and phenolic content in most experiments [31]. In addition, pineapple vinegar and pineapple vinegar mixed with dragon fruit without peel had higher acetic acid contents than commercial pineapple vinegar.

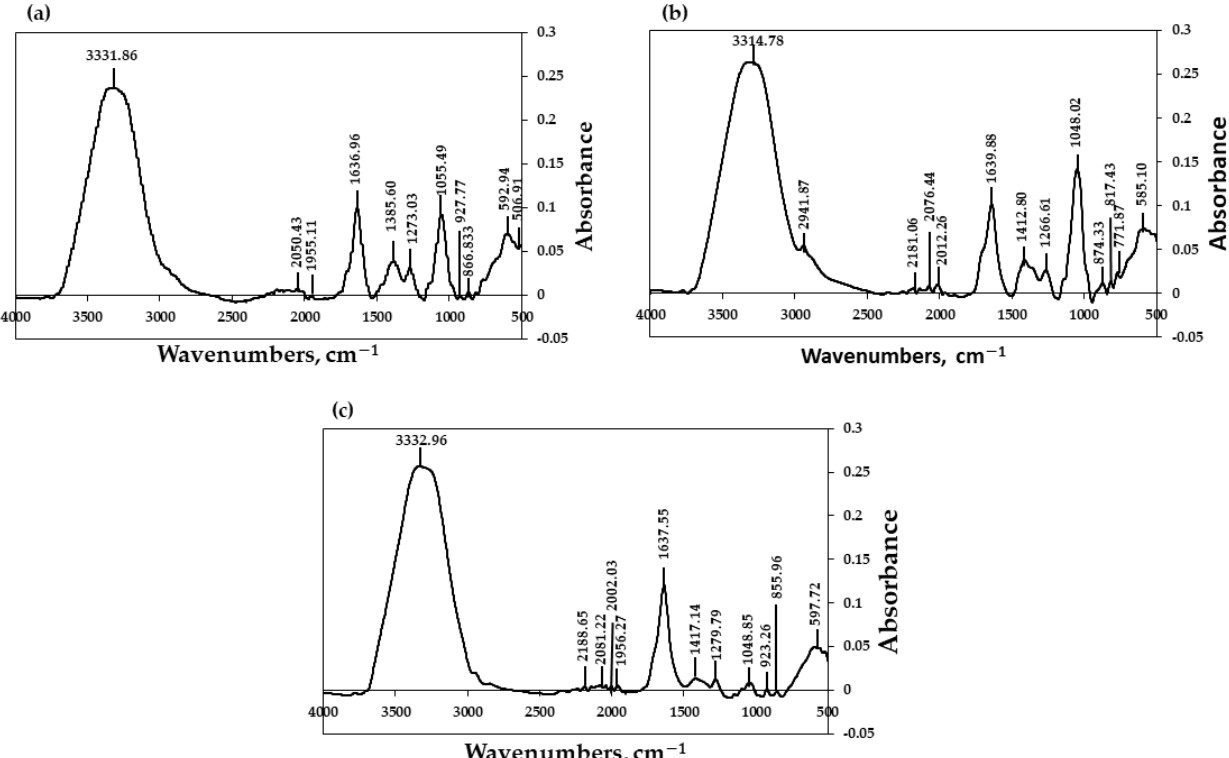

**Figure 4.** Fourier-transform infrared spectroscopy (FTIR) spectra of pineapple vinegar (**a**), pineapple vinegar mixed with red dragon fruit without peel (**b**), and pineapple vinegar mixed with red dragon fruit with peel (**c**).

## 4. Conclusions

In this study, the fermentation of pineapple vinegar and vinegar mixed with dragon fruit juice using SCF could increase the acetic acid content by more than 6% and the residual alcohol content by not more than 1.5%, thereby meeting the EU standard for wine vinegar using. Notably, this method required 20 days to convert ethanol to acetic acid. The addition of red dragon fruit to pineapple vinegar did not increase the total phenolic acid content or antioxidant activity. However, the addition of red dragon fruit with pulp and peel increased the antioxidant activity; however, the residual alcohol content exceeded the specified standard.

The addition of red dragon fruit to pineapple vinegar resulted in colors that will be attractive to consumers. Further, the total phenolic content of the fermented vinegar obtained using SCF was higher than that of commercial pineapple vinegar from a local market. This research shows the potential of local fruits to be developed as functional drinks with economic advantages and high contents of beneficial substances as well. This could be one method to develop functional food from local fruits.

**Author Contributions:** Conceptualization, A.B.; Data curation, C.T. and P.J.; Formal analysis, C.T. and P.J.; Investigation, S.J.; Project administration, S.K. and P.V.; Resources, K.N. and W.A.; Software, K.N. and W.A.; Supervision, N.N.; Validation, K.N. and W.A.; Writing—original draft, A.B.; Writing—review & editing, N.N. All authors have read and agreed to the published version of the manuscript.

**Funding:** This research was supported by the Kasetsart University Research and Development Institute, Thailand, grant number Kurdi (FF (KU) 15.64).

**Institutional Review Board Statement:** Not applicable.

**Informed Consent Statement:** Not applicable.

**Data Availability Statement:** All data supporting the conclusions of this article are included in the manuscript.

**Acknowledgments:** We thank the Kasetsart University Research and Development Institute (KURDI) and Agro-Industrial Product Improvement Institute, Kasetsart University, Thailand, for their support. This research was supported by the Research and Development Institute, Nakhon Sawan Rajabhat University (RDI).

**Conflicts of Interest:** The authors declare no conflict of interest.

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
