# Peer review of "Comparison of the Chemical Properties of Pineapple Vinegar and Mixed Pineapple and Dragon Fruit Vinegar"

_fermentation, doi:10.3390/fermentation8110597_

Round 1

Reviewer 1 Report (New Reviewer)

This work is well designed, presented and discussed

-Why the authors did not investigate the fermentation of pineapple juice with Dragon peel juice only. This should be conducted based on the decreasing the final cost of end product.

L148 Expand TPC

L315-319 should be moved to Table 4 as title and footnote and clarify match at Table 4??

Author Response

Reviewer 2 Report (New Reviewer)

The subject investigated is very interesting and gives opportunities for future use of pineapple and dragon fruit to produce vinegar. There are just 2 issues that I questioned and need to be improved:

 1.       It is not clear how were the vinegar fermentations performed? What kind of deposits were used? Were there replicas?

It seems essential to have duplicates or triplicates, but it is not clear if this was done. For small fermentation volumes, there are significant variation between replicas, therefore it is necessary to have replicates to increase the reliability of the results.

 2.       In “Materials and Methods” there is reference to the analysis of viability of microorganisms by colony count. Are the results shown? Was the viability followed through all fermentation? It would be very interesting to see how.

Author Response

This manuscript is a resubmission of an earlier submission. The following is a list of the peer review reports and author responses from that submission.

Round 1

Reviewer 1 Report

The manuscript's topic seems interesting, the chemical characterization of vinegar from pineapple and vinegar mixed from pineapple and dragon fruits. However, it needs significant language review and more work on the logical schema of results presentation, e.g., "Comparison of the chemical properties of pineapple vinegar and mixed pineapple and dragon fruit vinegar". I propose a transformation title: "Comparison of the chemical properties of vinegar from pineapple, and vinegar with the juice of dragon fruit".

Abstract: The schema of the abstract should contain a brief description of the aim of the research, methods, results with numerical findings and a short conclusion.

Moreover, "keywords" are so unspecific that it will be difficult to search for this article. 

Line 52-80: this big part of text has no reference.

The same line 134-139, no reference for mentioned methods 1) ethanol determination, 2) microorganisms counting, 3) acetic acid determination, 4) color of fruit.

L 179: chemical standards should be listed in the separate paragraph "Chemicals"

Table 2 no statistics.

Also no statists in Table 3, moreover, why only results are presented for day 0 and 20?

Table 4: only 6 compounds have been identified in the vinegars?  Volatile compounds in red wine vinegars obtained by submerged and surface acetification in different woods - ScienceDirect

and there is no value results for volatiles? only identification 

Author Response

We thank for your advice and suggestions, Please see the attachment

Reviewer 2 Report

The manuscript “Comparison of the chemical properties of pineapple vinegar and mixed pineapple and dragon fruit vinegar” is overall with scientific significance and interesting results. The results are rational and contain some interesting information that could be useful in making advances in developing novel functional vinegar products. The manuscript can be improved on some specific editing which are listed below:

1.       Abstract: Please specify what is “degraded pineapples”?

2.       Abstract: it difficult to understand “the addition of dragon fruit juice to the peel” … Please revise this sentence.

3.       Introduction is too long, and some contents are not concentrated on the topic. I suggest reducing size of the introduction and remove some basic information, for example, introduction of vinegar in the third paragraph.

4.       Legends under Figure 1 and Figure 2: please supplement the missing information after pineapple wine.

5.       How to prepare the dragon fruit peel juice? Is composition of the juice stable and repeatable by each preparation?

6.       Although addition of dragon fruit peel increases overall antioxidant activity, is it contains residue or insoluble fragments?

7.       Conclusion: “Owing to the high concentration of acetic acid in vinegar, it is considered a functional food with many benefits, such as the control of blood sugar and blood pressure.” This is a predication which is not appropriate in the conclusion section.

Author Response

(The authors gave the same response as above.)

Reviewer 3 Report

The work is interesting and can be accepted.

Author Response

We thank for your comment 

The Authors

Round 2

Reviewer 1 Report

Still, the presentation of GC-MS results is not acceptable. Have authors presented peak areas, concentrations or % of compounds' contribution in the total amount of volatiles? The information in table 4 is useless. 

Author Response

Comment from reviewer

Still, the presentation of GC-MS results is not acceptable. Have authors presented peak areas, concentrations or % of compounds' contribution in the total amount of volatiles? The information in table 4 is useless.

-As your suggestion, we have already added % peak areas.

Reviewer 2 Report

The authors have done extensive revisions. I have no more comments.